# Complex regulation of Ca$_v$2.2 N-type Ca$^{2+}$ channels by Ca$^{2+}$ and G-proteins

**Jessica R. Thomas[1], Jinglang Sun[2], Juan De la Rosa Vazquez[2], Amy Lee[2]\***

**1** Dept. of Biomedical Sciences, Meharry Medical College, Nashville, TN, United States of America, **2** Dept. of Neuroscience and Center for Learning and Memory, University of Texas-Austin, Austin, TX, United States of America

\* amy.lee1@austin.utexas.edu

**Data Availability Statement:** All relevant data are contained within the manuscript and its Supporting Information files.

**Funding:** This work was supported by the National Institutes of Health (https://www.nih.gov) under

## Abstract

G-protein coupled receptors inhibit Ca$_v$2.2 N-type Ca$^{2+}$ channels by a fast, voltage-dependent pathway mediated by Gα$_i$/Gβγ and a slow, voltage-independent pathway mediated by Gα$_q$-dependent reductions in phosphatidylinositol 4,5-bisphosphate (PIP2) or increases in arachidonic acid. Studies of these forms of regulation generally employ Ba$^{2+}$ as the permeant ion, despite that Ca$^{2+}$-dependent pathways may impinge upon G-protein modulation. To address this possibility, we compared tonic G-protein inhibition of currents carried by Ba$^{2+}$ ($I_{Ba}$) and Ca$^{2+}$ ($I_{Ca}$) in HEK293T cells transfected with Ca$_v$2.2. Both $I_{Ba}$ and $I_{Ca}$ exhibited voltage-dependent facilitation (VDF), consistent with Gβγ unbinding from the channel. Compared to that for $I_{Ba}$, VDF of $I_{Ca}$ was less sensitive to an inhibitor of Gα proteins (GDP-β-S) and an inhibitor of Gβγ (C-terminal construct of G-protein coupled receptor kinase 2). While insensitive to high intracellular Ca$^{2+}$ buffering, VDF of $I_{Ca}$ that remained in GDP-β-S was blunted by reductions in PIP2. We propose that when G-proteins are inhibited, Ca$^{2+}$ influx through Ca$_v$2.2 promotes a form of VDF that involves PIP2. Our results highlight the complexity whereby Ca$_v$2.2 channels integrate G-protein signaling pathways, which may enrich the information encoding potential of chemical synapses in the nervous system.

## Introduction

In nerve terminals, voltage-gated Ca$_v$2 channels are prominent mediators of Ca$^{2+}$ influx which triggers the exocytotic release of neurotransmitters into the synaptic cleft. The inhibition of presynaptic Ca$_v$2 channels by neurochemicals such as GABA and norepinephrine potently suppresses neurotransmission via receptors coupled to heterotrimeric G-proteins (GPCRs) [1]. This inhibition can occur through a voltage-dependent, membrane-delimited pathway involving the Gα$_{i/o}$ class of G-proteins and the binding of Gβγ to the channel [2–6]. GPCRs coupled to the Gα$_q$ class of G-proteins also inhibit Ca$_v$2 channels through a slower, voltage-independent pathway [7, 8]. Mechanisms for this form of Ca$_v$2 channel modulation include enzymatic depletion of phosphatidylinositol 4,5-bisphosphate (PIP2) [9, 10], which normally enhances the activity of Ca$_v$ channels [11].

grants R01 EY026817 and R03TR005086 and startup funds from The University of Texas at Austin (to A.L.). The sponsors did not play a role n the study design, data collection and analysis, decision to publish, or preparation of the manuscript.

**Competing interests:** The authors have declared that no competing interests exist.

Among the $Ca_v2$ subtypes ($Ca_v2.1$, $Ca_v2.2$, $Ca_v2.3$), $Ca_v2.2$ channels exhibit particularly strong voltage-dependent inhibition by G-proteins [12, 13] which can be tempered by other signal mediators. For example, GPCRs that activate protein kinase C (PKC) diminish the impact of Gβγ on $Ca_v2.2$ [14–16]. PKC phosphorylates a threonine in the cytoplasmic linker between domains I and II, which prevents the interaction with Gβγ [17, 18]. Conversely, several proteins involved with synaptic release, such as syntaxin 1A and cysteine string proteins, enhance G-protein inhibition of $Ca_v2.2$ through interactions with both Gβγ and the channel [19–21]. Thus, the impact of GPCRs on neuronal $Ca_v2.2$ channels may vary with patterns of neuronal activity, exposure to various neuromodulators, and interactions with proteins in specific subcellular compartments.

$Ca_v2.2$ undergoes some voltage-dependent inhibition by G-proteins even without exogenous application of GPCR agonists, which could result from an excess of free Gβγ and/or activation of autoreceptor GPCRs [22–27]. In these studies, $Ba^{2+}$ was often chosen as the permeant ion since $Ba^{2+}$ currents ($I_{Ba}$) are larger in amplitude than $Ca^{2+}$ currents ($I_{Ca}$) [28]. However, this approach can mask physiologically relevant forms of $Ca_v2.2$ modulation that rely on $Ca^{2+}$ influx [29] and could affect the impact of G-proteins. To address this possibility, we compared tonic G-protein modulation of $I_{Ba}$ and $I_{Ca}$ in HEK293T cells transfected with $Ca_v2.2$. Our results indicate that tonic inhibition by Gβγ is stronger for $I_{Ba}$ than for $I_{Ca}$ and implicate PIP2 in modulation of $I_{Ca}$ and not $I_{Ba}$ when Gβγ-mediated inhibition is suppressed. Our findings add to the diverse modes by which $Ca_v$ channels are regulated, some of which depend critically on the nature of the permeating cation.

## Materials and methods

*cDNAs.* The following cDNAs were used: $Ca_v2.2$ e37b (Genbank # AF055477), $β_{2a}$ (Genbank # NM_053851), $α_2δ$-1 (Genbank # NM_000722.3), pEGFP (Addgene). The C-terminal construct corresponding to G-protein coupled receptor kinase containing a myristic acid attachment signal (MAS-GRK2-ct) and zebrafish voltage-sensitive phosphatase (Dr-VSP) were described previously [8, 11].

### Cell culture and transfection

Human embryonic kidney 293 cells transformed with the SV40 T-antigen (HEK 293T, American Type Culture Collection Cat# CRL-3216, RRID:CVCL_0063) were maintained in Dulbecco's modified Eagle's medium with 10% fetal bovine serum and 1% penicillin-streptomycin at 37°C in a humidified atmosphere with 5% $CO_2$. Cells were grown to 80% confluence and transfected using Fugene 6 (Promega) according to the manufacturer's protocol. Cells were plated in 35 mm dishes and transfected with cDNAs encoding $Ca_v$ channel subunits ($Ca_v2.2$, 1.8 μg; $β_{2a}$, 0.6 μg; and $α_2δ$-1, 0.6μg). In some experiments, 0.5 μg of MAS-GRK2-ct or Dr-VSP was co-transfected to buffer Gβγ or deplete PIP2, respectively. Cotransfection with cDNA encoding enhanced green fluorescent protein (pEGFP, 50 ng) allowed visualization of transfected cells.

### Electrophysiological recordings

Whole-cell patch recordings were performed 24–72 hours after transfection with a EPC-8patch clamp amplifier and Patch master software (HEKA Elektronik). External recoding solution contained (in mM): 150 Tris, 1 $MgCl_2$, and 5 $CaCl_2$ or $BaCl_2$. Intracellular solution contained (in mM): 140 N-methyl-D-glucamine 10 HEPES, 10 EGTA, 2 $MgCl_2$, and 2 Mg-ATP. The pH of both solutions was adjusted to 7.3 using methanesulfonic acid. In some experiments BAPTA or Guanosine5′-[β-thio]diphosphate trilithium salt (GDPβS) was added to the

intracellular solution to either buffer Ca$^{2+}$ or block G proteins, respectively. Electrode resistances were 4–6 MΩ in the bath solution. Series resistance was compensated 60–70%. Leak currents were subtracted using a P/-4 protocol. Data were analyzed using Igor Pro software (Wavemetrics). Averaged data represent mean ± S.E., and result from at least 3 independent transfections.

### Data presentation and statistical analysis

Data were incorporated into figures using Graph-Pad Prism software and Adobe Illustrator software. Statistical analysis was performed with Graph-Pad Prism software. The data were first analyzed for normality using the Shapiro–Wilk test. For parametric data, significant differences were determined by Student's t test or ANOVA with post hoc Dunnett or Tukey test. For nonparametric data, the Mann-Whitney, Kruskal–Wallis, or Wilcoxon tests were used as well as post hoc Dunn's test.

## Results

In electrophysiological recordings of Ca$_v$ channels, inhibition by G-proteins can be studied by evoking current-voltage (I-V) relationships before (P1) and after (P2) a depolarizing prepulse [23]. With this protocol, current amplitudes after the prepulse should be larger due to Gβγ unbinding from the channel [6]. We used this voltage protocol to test whether the tonic Ca$_v$2.2 modulation by G-proteins might differ for $I_{Ba}$ and $I_{Ca}$ in transfected HEK293T cells. In our experiments, we used the Ca$_v$2.2 splice variant containing exon 37b which lacks the voltage-independent, tyrosine kinase-dependent form of G-protein modulation seen for variants containing exon 37a [30]. We cotransfected cells with the auxiliary α$_2$δ-1 subunit and β$_{2a}$ subunit, which produces stronger tonic G-protein modulation than channels containing the β$_{1b}$ subunit [23]. To account for differences in current amplitudes between cells due to variable levels of channel expression, we plotted I-V data normalized to the maximal current evoked by P2 ($I_{norm}$). As expected, the amplitudes of the normalized peak Ba$^{2+}$ current ($I_{norm}$, at test pulse = 0 mV) were significantly higher after (median = -0.84) than before a +60-mV prepulse (median = -0.33, W = -28, p = 0.02 by Wilcoxon matched-pairs test; Fig 1A–1C). To verify the involvement of G-proteins, we used the guanosine diphosphate analog GDP-β-S which should limit the availability of Gβγ by stabilizing its association with Gα [31]. When GDP-β-S was included in the intracellular recording solution, the amplitude of peak $I_{norm}$ was still higher after (median = -0.58) than before the prepulse (median = -0.41, W = -28, p = 0.02 by Wilcoxon matched-pairs test; Fig 1D–1F). However, the extent of the prepulse-induced increase in peak $I_{norm}$ was 10-fold lower with GDP-β-S (Fractional facilitation (FF) = 1.05 ± 0.27 for control vs. 0.2 ± 0.04 for +GDP-β-S, t = 3.081, df = 12, p = 0.01 by unpaired t-test; Fig 1G). These results show that Ca$_v$2.2 undergoes tonic, voltage-dependent inhibition of Ca$_v$2.2 by G-proteins in HEK293T cells, as described previously for this channel in other cell-types [23, 26].

Like $I_{Ba}$, $I_{Ca}$ also was increased by the prepulse under control conditions (mean = -0.56 before vs. mean = -0.74 after, t = 6.348, df = 6, p = 0.001; Fig 2A–2C) and with GDP-β-S (median = -0.68 before vs. median = -0.82 after, W = -66, p = 0.001 by Wilcoxon matched-pairs test; Fig 2D–2F). However, the depolarizing prepulse caused a significantly smaller increase in $I_{Ca}$ than $I_{Ba}$ under control conditions (FF = 0.35 ± 0.03 for $I_{Ca}$ vs. 1.05 ± 0.27 for $I_{Ba}$, t = 2.549, df = 12, p = 0.02 by unpaired t-test). Moreover, facilitation caused by the prepulse did not significantly differ under control conditions and with GDP-β-S (FF = 0.25 ± 0.03, t = 2.007, df = 16, p = 0.06 compared to control by unpaired t-test; Fig 2G). Thus, tonic voltage-dependent inhibition of Ca$_v$2.2 by G-proteins is weaker and less sensitive to GDP-β-S for $I_{Ca}$ than $I_{Ba}$.

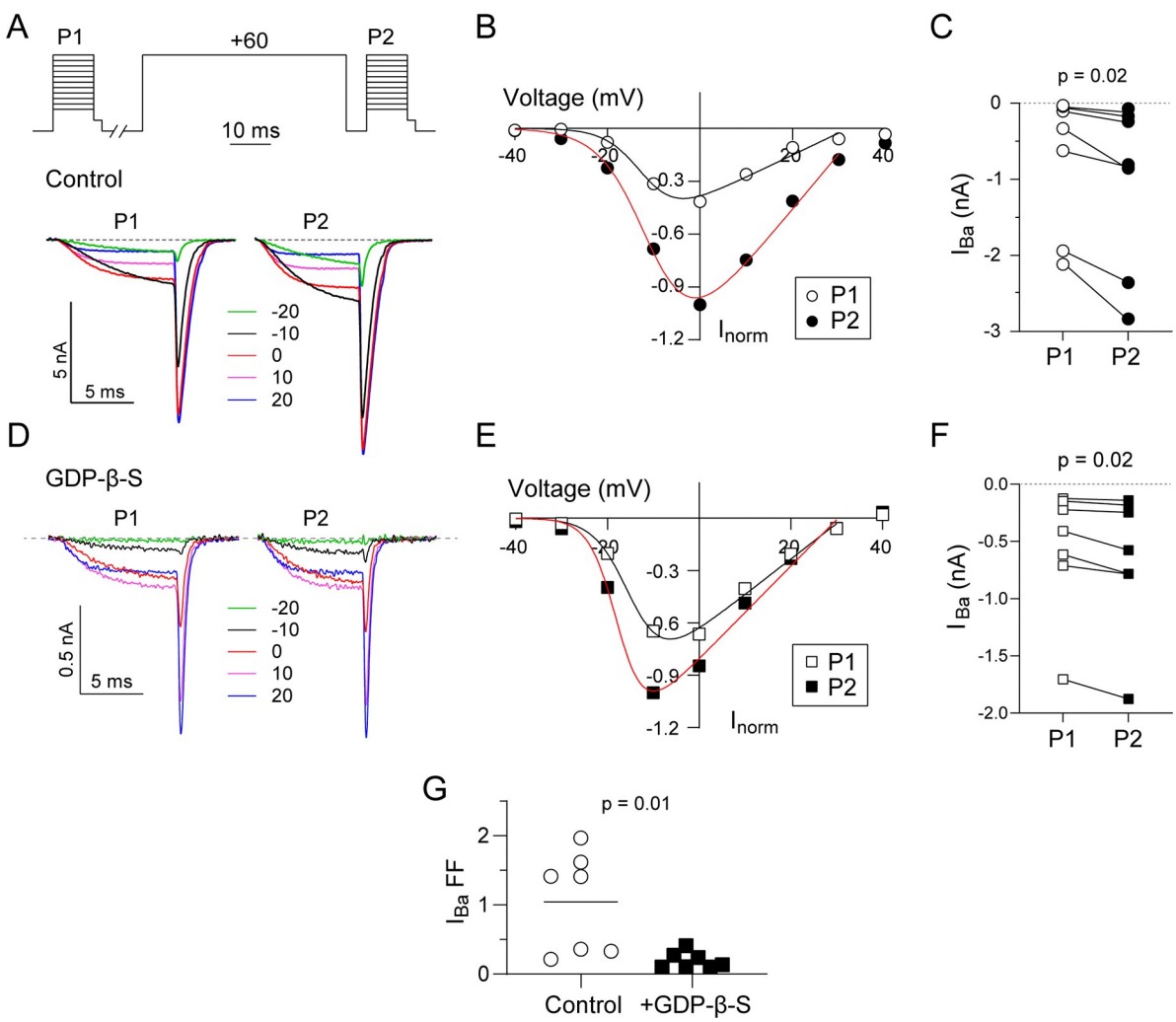

**Fig 1. VDF of I$_{Ba}$ for Ca$_v$2.2 is blunted by GDPβS.** (A) Representative current traces and voltage protocol. I$_{Ba}$ was evoked by a 10-ms test pulse from a holding voltage of -80 mV to the indicated voltages 10 s before (P1) and 5 ms after (P2) a 50-ms conditioning pre-pulse to +60 mV. The test pulses were followed by a 2-ms step to -60 mV prior to repolarizing to -80 mV. (B) Representative I-V plot of P1 and P2 currents for a single cell. I$_{norm}$ represents the amplitude of the steady state current near the end of P1 or P2 pulse normalized to the maximal current evoked by the P2 voltage. Smooth line represents Boltzmann fits. (C) I$_{Ba}$ for P1 and P2 pulses (both 0 mV) for each cell. (D-F) Same as in A-C but for cells where GDPβS (0.3 mM) was included in the intracellular recording solution. (G) Plot comparing fractional facilitations, (P2-P1)/P1, for I$_{Ba}$ evoked by 0 mV test pulse between cells with and without intracellular GDPβS. Bar represents mean. p-value was determined by Wilcoxon test (C, F) or unpaired t-test (G).

To further investigate this difference in G-protein regulation of $I_{Ca}$ and $I_{Ba}$, we used a double pulse protocol where the effect of varying the voltage of the prepulse is measured on a test current evoked before (P1) and after (P2) the prepulse (Fig 3A–3F). With this protocol, VDF is evident as a progressive increase in the P2 vs P1 current amplitude with prepulse voltage [32]. For $I_{Ba}$, VDF was robust under control conditions and was reduced by GDP-β-S (Fig 3B and 3D). The amount of VDF was measured as the difference in the P2 and P1 currents with a prepulse to +80 mV (Fractional facilitation, FF$_{80}$) and was significantly lower with GDP-β-S (mean FF$_{80}$ = 0.13) compared to control conditions (mean FF$_{80}$ = 0.48, t = 3.748, df = 14, p = 0.0022 by unpaired t-test; Fig 3F). VDF of $I_{Ca}$ was also strong and showed a similar dependence on prepulse voltage as $I_{Ba}$. However, unlike $I_{Ba}$, VDF was not significantly different

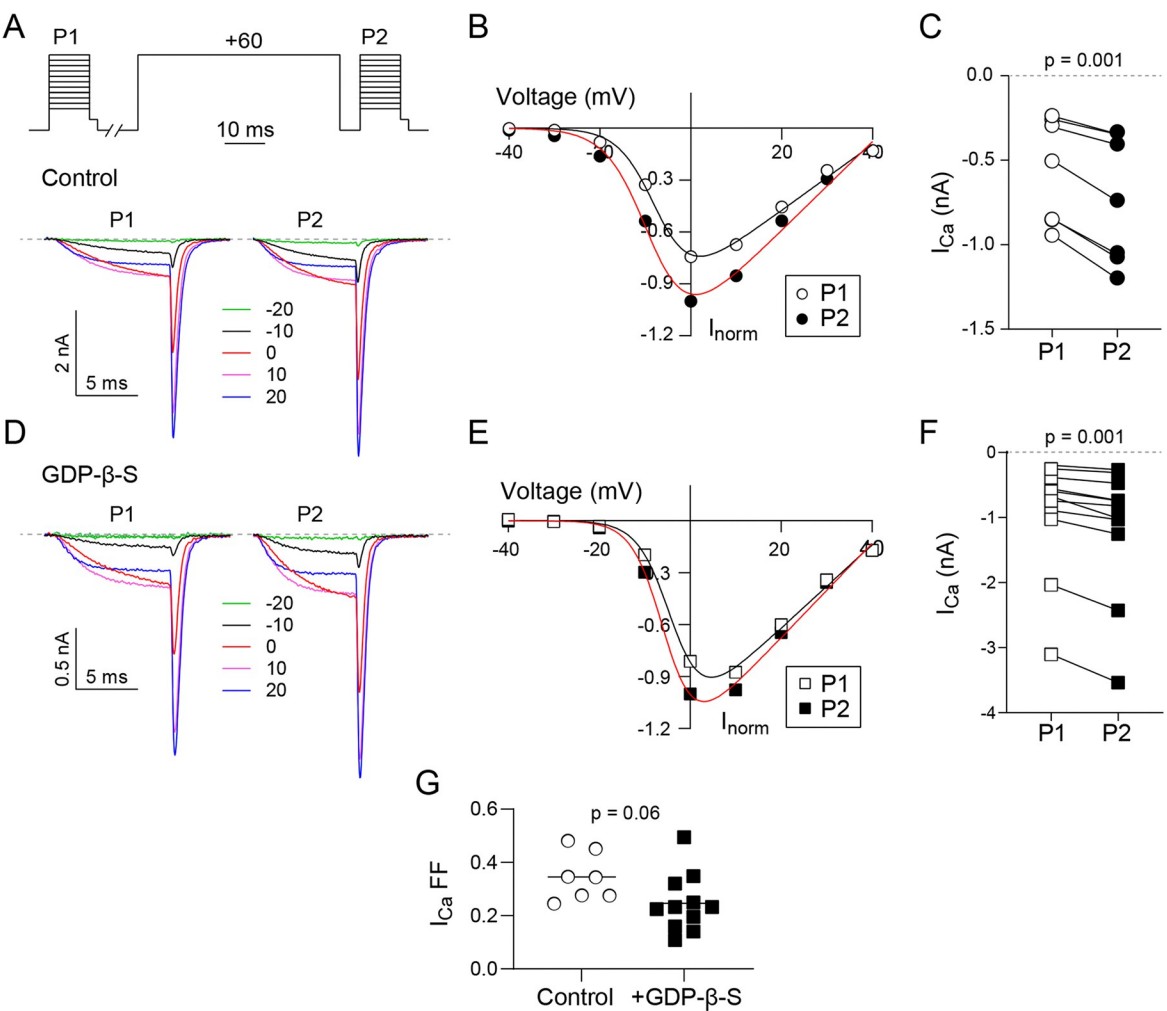

**Fig 2. VDF of $I_{Ca}$ for $Ca_v2.2$ is unaffected by GDPβS.** (A-G) Same voltage protocol and analysis as in Fig 1A and 1B) Representative current traces and voltage protocol (A) and I-V plot (B) for a single cell. Smooth line represents Boltzmann fits. (C) $I_{Ca}$ for P1 and P2 pulses (both at 0 mV) for each cell. p-value was determined by paired t-test. (D-F) Same as in A-C but for cells where GDPβS (0.3 mM) was included in the intracellular recording solution. (G) Plot comparing fractional facilitations, (P2-P1)/P1, for $I_{Ca}$ evoked by 0 mV test pulse between cells with and without intracellular GDPβS. Bar represents mean. p-value was determined by paired t-test (C), Wilcoxon test (F) or unpaired t-test (G).

under control conditions (mean $FF_{80}$ = 0.52) and with GDP-β-S (mean $FF_{80}$ = 0.39, t = 2.076, df = 15, p = 0.056 by unpaired t-test; Fig 3C, 3E and 3F).

A possible explanation for our results thus far was that VDF of $I_{Ca}$ could involve an additional pathway that is recruited even when Gβγ is inhibited. To test this, we coexpressed $Ca_v2.2$ with a C-terminal construct of GPCR kinase 2 (GRK) which has no kinase activity but acts to sequester Gβγ [8]. With the I-V protocol to measure VDF, the peak current amplitude for both $I_{Ca}$ and $I_{Ba}$ was still increased by the +60 mV conditioning pulse in the presence of GRK (Fig 4A–4F). As expected, VDF for $I_{Ba}$ was significantly weaker with GRK (FF = 0.19 ± 0.04, n = 7) than under control conditions (FF = 1.41 ± 0.27, n = 7; t = 3.102, df = 12, p = 0.009 by unpaired t-test; Fig 4G). In contrast, there was no significant difference in VDF for $I_{Ca}$ with GRK (FF = 0.32 ± 0.07, n = 5) than under control conditions (FF = 0.34 ± 0.03, n = 7; t = 0.314, df = 10, p = 0.759 by unpaired t-test; Fig 4G). Similar results were obtained with the double pulse protocol (Fig 5A–5D). Compared to control conditions,

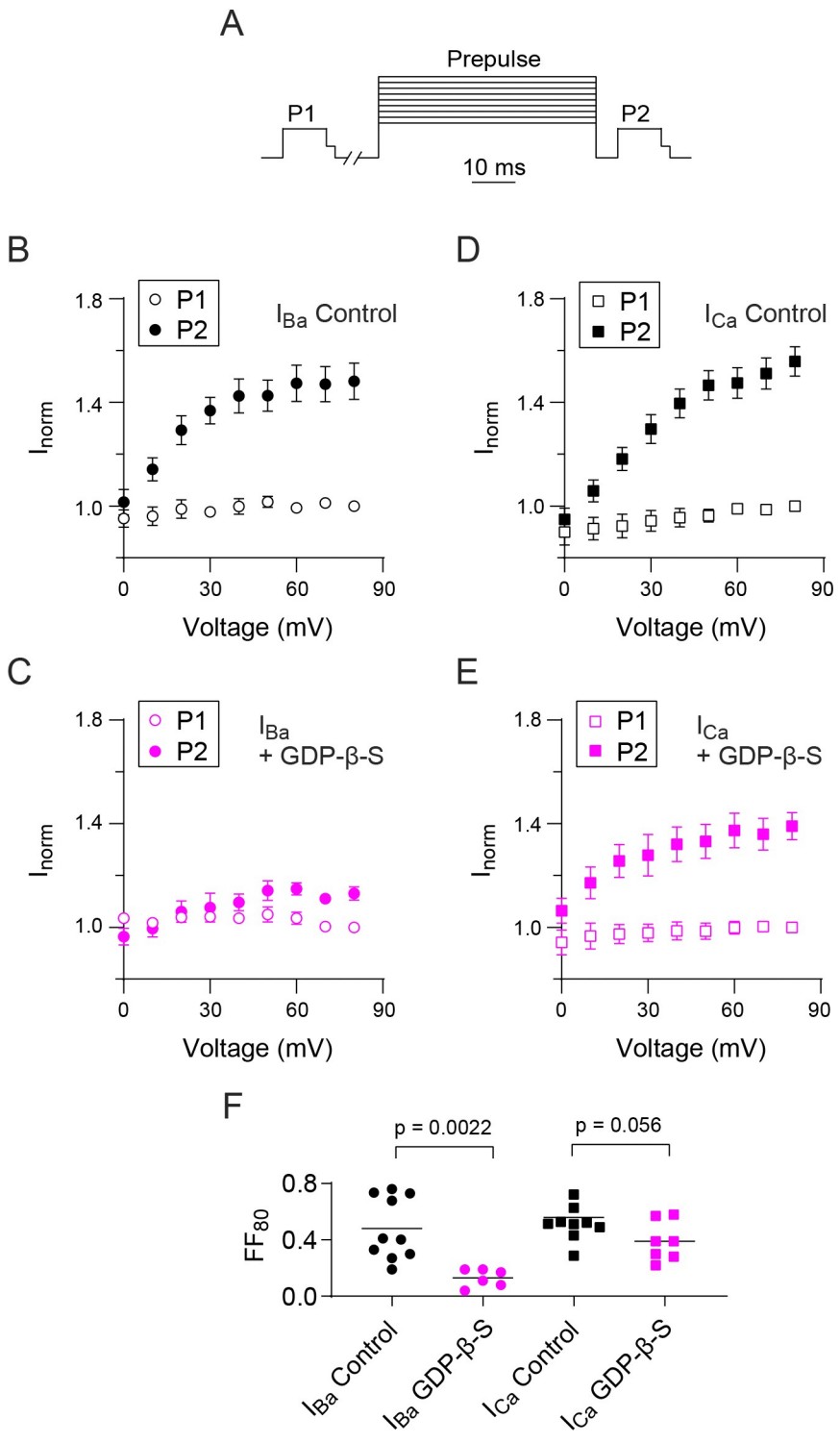

**Fig 3. Decline of VDF of I$_{Ba}$ for Ca$_v$2.2 caused by GDPβS.** (A) Voltage protocol. I$_{Ca}$ (or I$_{Ba}$) was evoked by a 10-ms test pulse from a holding voltage of -80 mV to -5 mV (-10 mV for I$_{Ba}$) 10-s before (P1) and 5-ms after (P2) a 50-ms conditioning pre-pulse to indicated voltages. The test pulses were followed by a 2-ms step to -60 mV prior to repolarizing to -80 mV to facilitate measurements of tail currents. (B, C) Tail currents for I$_{Ca}$ or I$_{Ba}$ evoked by P1 or P2 test pulses were normalized to that for the P1 pulse prior to the +80 mV prepulse (I$_{norm}$) and plotted against the prepulse voltage. (D, E) Same as in B-C but for cells where GDPβS (0.3 mM) was included in the intracellular recording solution. (F) Plot comparing fractional facilitations, (P2-P1)/P1, for I$_{Ca}$ and I$_{Ba}$ evoked before and after a +80

mV conditioning prepulse in cells with and without intracellular GDPβS. Bars represent mean. p-values were determined by unpaired t-test.

GRK expression caused a significant reduction in VDF for $I_{Ba}$ (58%, Fig 5B and 5D) but a non-significant slight increase in VDF for $I_{Ca}$ (Fig 5C and 5D). These results agree with our hypothesis that VDF of $I_{Ca}$ could proceed even when Gβγ is inhibited.

The apparent absence of an effect of GDPβS on VDF of $I_{Ca}$ (Figs 2 and 3) could signify opposing regulation of Ca$_v$2.2 by another G-protein signaling pathway that is recruited when Ca$^{2+}$ ions permeate the channel. One possibility was that GDP-β-S enabled a form of Ca$^{2+}$-dependent facilitation (CDF) similar to that for Ca$_v$2.1 channels that is mediated by calmodulin (CaM) binding to the Ca$_v$2.1 C-terminal domain [33, 34]. This seemed unlikely since the VDF exhibited by Ca$_v$2.2 $I_{Ca}$ in the double pulse protocol did not resemble CaM-dependent CDF of Ca$_v$2.1, which shows a U-shaped dependence on prepulse voltage that reflects the amount of Ca$^{2+}$ influx during the

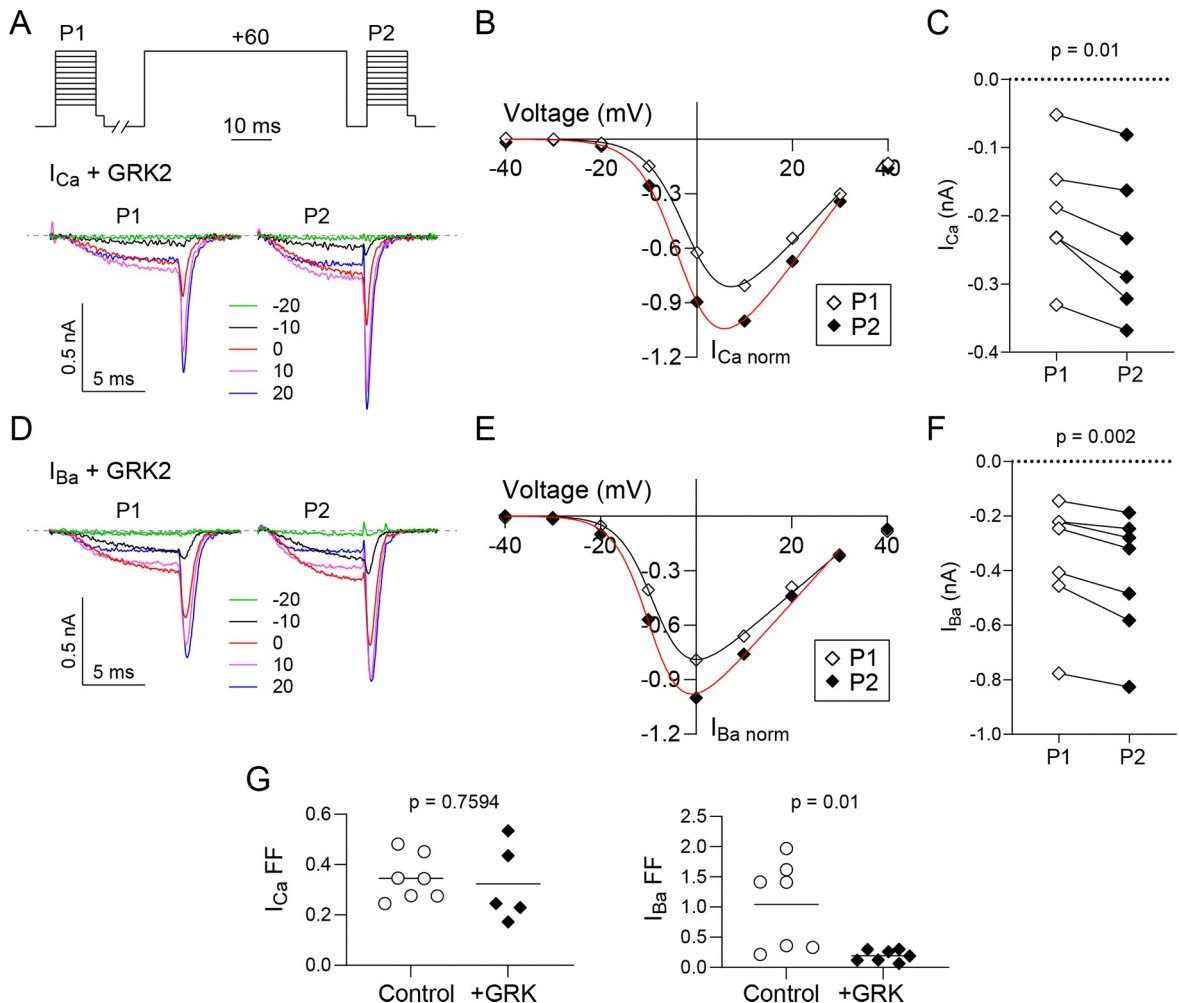

**Fig 4. VDF of I$_{Ba}$ for Ca$_v$2.2 is suppressed by GRK2.** (A) Representative current traces and voltage protocol. I$_{Ca}$ was evoked by the same voltage protocol as in Fig 1. (B) Representative I-V plot of P1 and P2 currents for a single cell co-transfected with GRK2. Smooth line represents Boltzmann fits. (C) I$_{Ca}$ for P1 and P2 pulses for each cell. p-value was determined by paired t-test. (D-F) Same as in A-C but for cells recorded in Ba$^{2+}$ bath solution. (G) Plots comparing fractional facilitations, (P2-P1)/P1, for I$_{Ca}$ and I$_{Ba}$ evoked by 0 mV test pulse. Bars represent mean. P-values were determined by unpaired t-test.

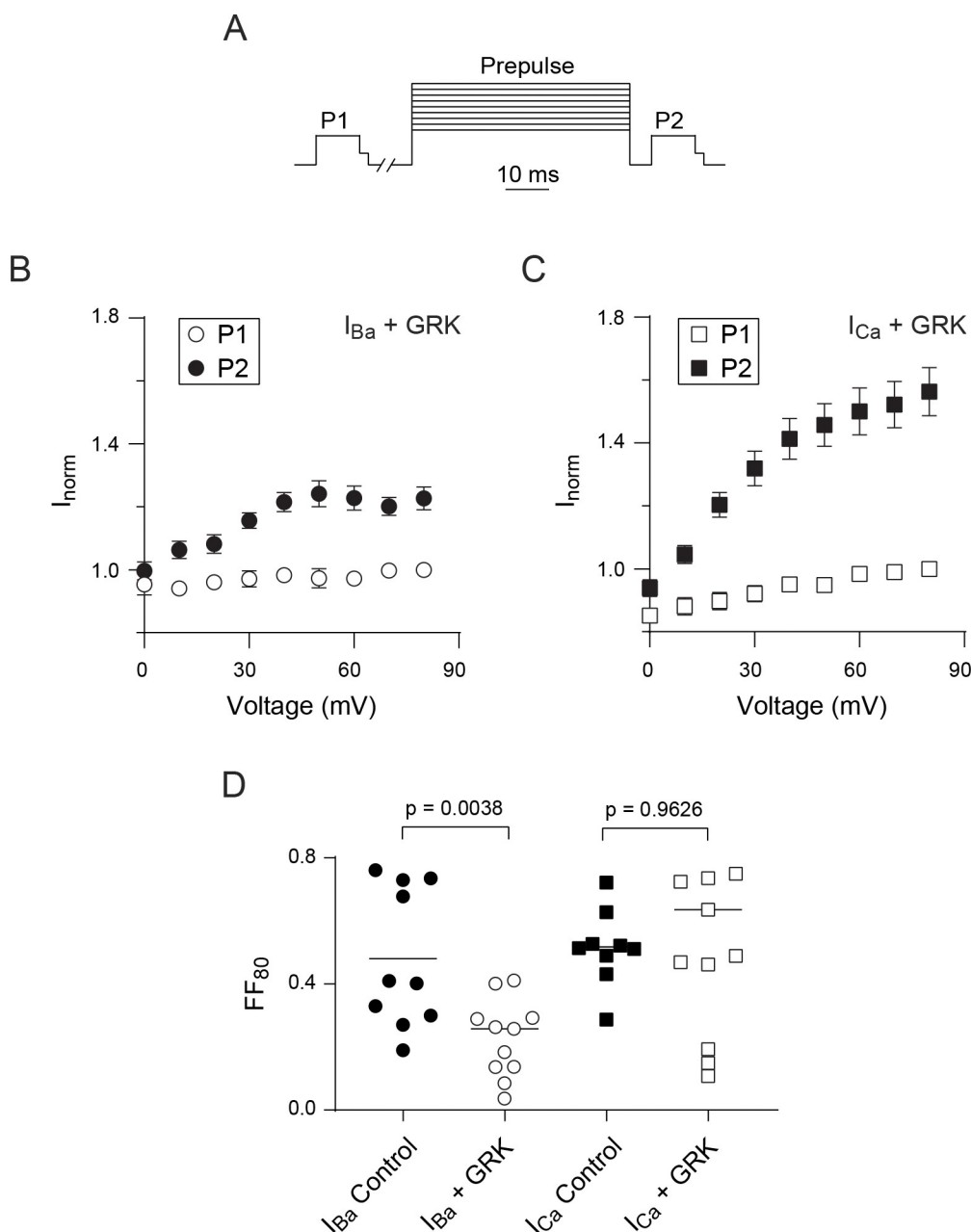

**Fig 5. Reduced VDF of I$_{Ba}$ for Ca$_v$2.2 caused by GRK2.** (A) Voltage protocol (same as in Fig 3). (B-C) Tail currents for I$_{Ca}$ or I$_{Ba}$ evoked by P1 or P2 test pulses were normalized to that for the P1 pulse prior to the +80 mV prepulse (I$_{norm}$) and plotted against the prepulse voltage. Cells were co-transfected with GRK2. (D) Plot comparing fractional facilitations, (P2-P1)/P1, for I$_{Ca}$ and I$_{Ba}$ evoked before and after an 80 mV conditioning pre pulse in cells with and without GRK2 transfection. Bars represent mean. p-values were determined by unpaired t-test.

prepulse [35, 36]. Moreover, Ca$_v$2.2 lacks key domains present in Ca$_v$2.1 that are required for CaM-dependent CDF [37]. In primary sensory neurons, Ca$_v$2.2 undergoes CDF that is mediated by CaM dependent protein kinase II (CaMKII), which requires cytoplasmic accumulation of Ca$^{2+}$ [38]. In the voltage protocols for measuring VDF, the interval between the P1 and conditioning pulses is 10 s, which may allow for sufficient Ca$^{2+}$ influx during the P1 pulse to activate Ca$^{2+}$-dependent pathways such as those involving CaMKII. However, VDF of $I_{Ca}$ with strong

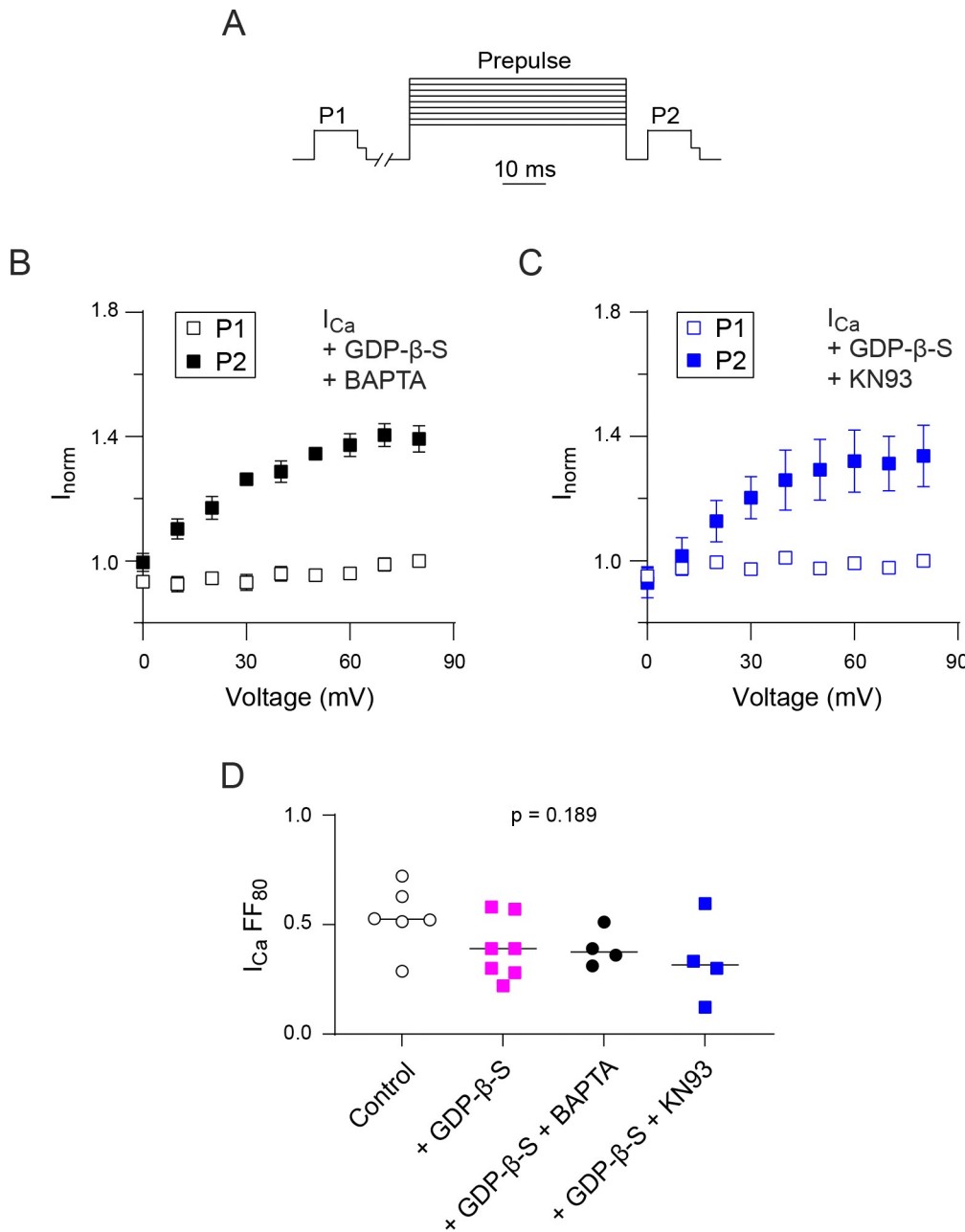

**Fig 6. VDF of I$_{Ca}$ for Ca$_v$2.2 is not mediated by CaMKII.** (A) Voltage protocol (same as in Fig 3). (B) Tail currents for I$_{Ca}$ evoked by P1 or P2 test pulses were normalized to that for the P1 pulse prior to the +80 mV prepulse (I$_{norm}$) and plotted against the prepulse voltage. GDPβS (0.3 mM) and BAPTA (10 mM) was included in the intracellular recording solution. (C) Same as in B but for cells where GDPβS (0.3 mM) and KN93 was included in the intracellular recording solution. (D) Plot comparing fractional facilitations, (P2-P1)/P1, for I$_{Ca}$ evoked before and after a +80 mV conditioning prepulse in cells with various intracellular conditions. Bars represent mean. p-value was determined by One-Way ANOVA.

Ca$^{2+}$ buffering with 10 mM BAPTA (FF = 0.39 ± 0.04, Fig 6A and 6B) or the 0.3 mM of the CaM-KII inhibitor KN93 (FF = 0.34 ± 0.10, Fig 6C) was similar to that under control conditions with (FF = 0.39 ± 0.05) or without GDP-β-S (FF = 0.53 ± 0.06; F(3, 17) = 1.781, p = 0.189 by One-Way ANOVA; Fig 6D). These results argue against a role for CaMKII in VDF of Ca$_v$2.2 I$_{Ca}$.

An alternative mechanism involves Gα$_q$-dependent activation of phospholipase C (PLC), which causes the hydrolysis of PIP2 into inositol 1,4,5-trisphosphate and diacylglycerol, or increased liberation of arachidonic acid by phospholipase A2 [39]. It is well-established that PIP2 supports the function of Ca$_v$ channels, and that GPCRs linked to Gα$_q$ cause a decline in PIP2 that lowers activity of Ca$_v$ channels [9, 11, 40, 41]. GDP-β-S might suppress this Gα$_q$ signaling pathway, thus increasing Ca$_v$ channel activity by reducing PIP2 hydrolysis. If selective for $I_{Ca}$, this effect of GDP-β-S on Gα$_q$ might mask the effect of GDP-β-S on the Gα$_{i/o}$/ Gβγ-mediated pathway, leaving net VDF unchanged. To test this, we utilized a voltage-sensitive phosphatase (VSP) from zebrafish which enables the depletion of PIP2 in living cells following a strong depolarizing voltage step (i.e., +120 mV). This approach has been used previously to blunt Gα$_q$-dependent inhibition of Ca$_v$ channels [11].

To enable VSP activation, we modified our voltage protocol to include a +120-mV VSP-activating pulse prior to the P1 test pulse and VDF was measured as the ratio of the P2/P1 pulses with an intervening +20 mV prepulse (Fig 7A). A more modest depolarizing prepulse was used in these experiments to avoid additional activation of the VSP. GDPβS was included in the intracellular recording solution to replicate conditions that led to distinctions in VDF of $I_{Ca}$ and $I_{Ba}$ in Fig 3. When the double pulse protocol was given without the +120-mV pulse, P2/P1 for $I_{Ca}$ did not differ in cells with (median = 1.371) and without VSP (median = 1.471; t = 1.612, df = 12, p = 0.133 by unpaired t-test) indicating that VSP did not affect VDF when not activated. In cells transfected with VSP, P2/P1 for $I_{Ca}$ measured with the +120 mV pulse (mean = 1.016 ± 0.04) was significantly lower than when measured without the +120 mV pulse (mean = 1.39 ± 0.046; t = 8.264, df = 7, p < 0.0001 by paired t-test; Fig 7B and 7F). This result demonstrates that PIP2 enhances VDF of $I_{Ca}$. As in control cells transfected with Ca$_v$2.2 alone (i.e., -VSP), P2/P1 for $I_{Ba}$ (+VSP) was not significantly different with (median = 1.184) or without the +120 mV pulse (median = 1.344, W = -15, p = 0.426 by Wilcoxon matched pair signed rank test; Fig 7D and 7F). Therefore, alterations in PIP2 do not affect VDF for $I_{Ba}$. Taken together, our results suggest that PIP2 enhances VDF of $I_{Ca}$ but not $I_{Ba}$ when G-proteins are inhibited.

## Discussion

Our study reveals an unusual feature of G-protein modulation of Ca$_v$2.2 that requires the influx of Ca$^{2+}$ through the channel. For $I_{Ba}$, VDF depends mainly on Gα$_{i/o}$/ Gβγ which is blunted by GDPβS (Figs 1, 3F and 8A) and GRK (Figs 4 and 5). For $I_{Ca}$, VDF that remains in the presence of GDPβS (Figs 2 and 3) requires PIP2 since it is suppressed by Dr-VSP (Fig 7A–7C). We propose that when Ca$^{2+}$ permeates the channel, GDPβS inhibits not only Gα$_{i/o}$/ Gβγ but also Gα$_q$/ Gβγ. The latter pathway promotes a decline in PIP2 since both Gα$_q$ and Gβγ can activate PLC [42]. Despite the competing effects of blunting the Gα$_{i/o}$/ Gβγ pathway, GDPβS strengthens VDF of $I_{Ca}$ by limiting Gα$_q$-dependent reductions in PIP2 (Fig 8B).

In studies of Ca$_v$ channels, Ba$^{2+}$ is often substituted for Ca$^{2+}$ in the extracellular recording solution in part to minimize Ca$^{2+}$-dependent pathways that could complicate analysis of intrinsic channel properties. However, $I_{Ca}$ can differ from $I_{Ba}$ in physiologically relevant ways. A prominent example is Ca$^{2+}$-dependent inactivation (CDI), which is a characteristic of all Ca$_v$1 and Ca$_v$2 channels and manifests as faster decay of $I_{Ca}$ compared to $I_{Ba}$ [36, 43]. Ca$_v$2 channels also undergo CDF [33–35], which for Ca$_v$2.2 requires CaMKII and is reduced following nerve injury [38]. The VDF of $I_{Ca}$ for Ca$_v$2.2 in our study differed from CaMKII-dependent CDF since it was not blocked by high BAPTA or the CaMKII inhibitor (Fig 6). The BAPTA-insensitivity suggests that Ca$^{2+}$ elevations within a nanodomain of the Ca$_v$2.2 channel are needed to support VDF when G-proteins are inhibited. Considering that Ca$^{2+}$ can increase

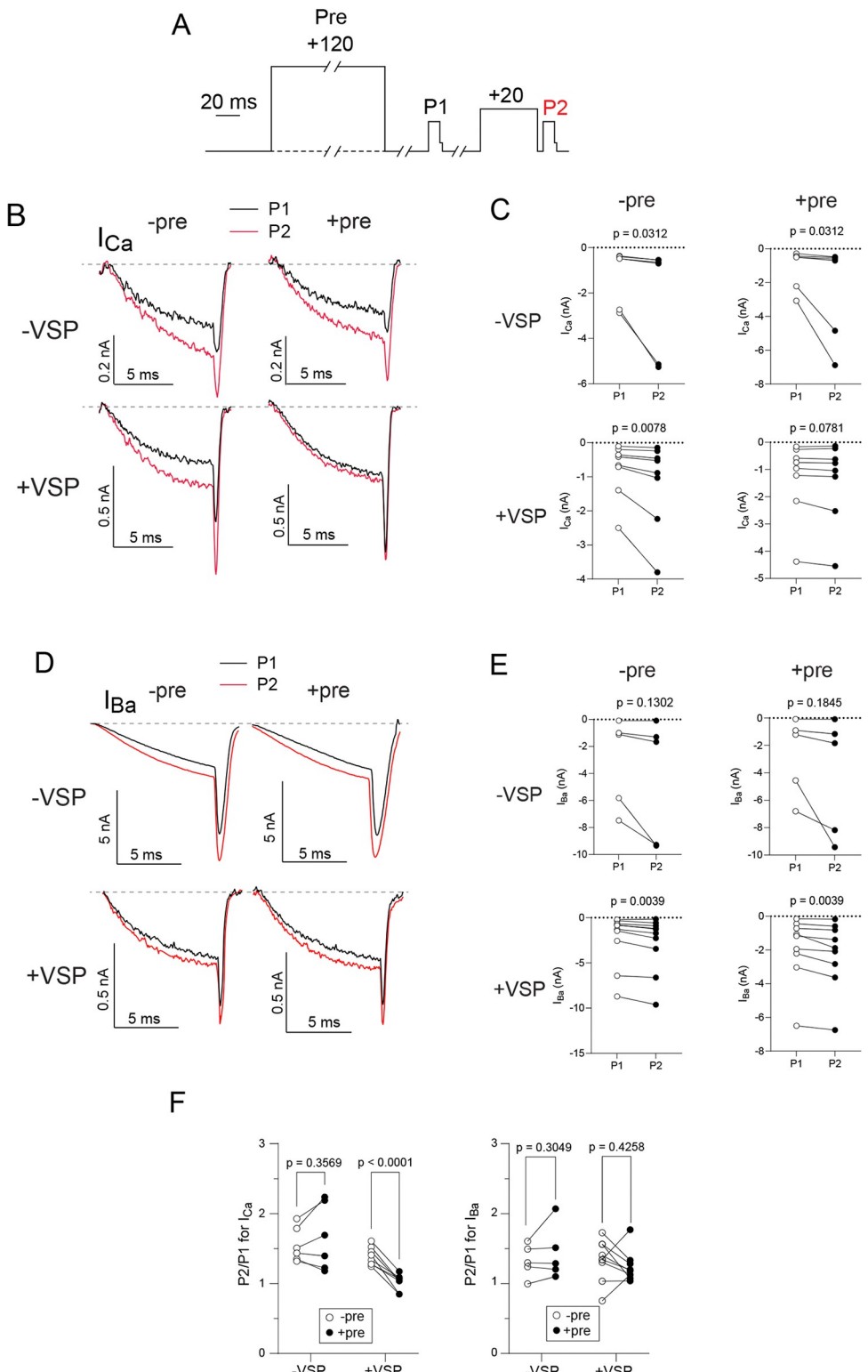

**Fig 7. VDF of I$_{Ca}$ for Ca$_v$2.2 is abolished after PIP2 depletion via VSP activation.** (A) Voltage protocol. An optional 1-s long +120 mV VSP activating pulse from a holding voltage of -80 mV is applied 10-s prior to the first test pulse. I$_{Ca}$ or I$_{Ba}$ was evoked by a 10-ms test pulse from a holding voltage of -80 mV to the indicated voltages (-5 mV for I$_{Ca}$ or -10 mV for I$_{Ba}$) 10-s before (P1) and 5-ms after (P2) a 50-ms conditioning pre-pulse to +20 mV. The test pulses were followed by a 2-ms step to -60 mV prior to repolarizing to -80 mV to facilitate measurement of the tail current. (B)

Paired representative I$_{Ca}$ traces reflecting VDF from VSP and non-VSP transfected cells, with each cell tested with and without a +120 mV VSP activating pulse. (C) P1 and P2 evoked I$_{Ca}$ comparison, obtained in conditions as dscribed in B. p-values determined by Wilcoxon test.(D, E) Same as in B-C but for cells recorded in Ba$^{2+}$ bath solution. p-values determined by paired t-test (-VSP) and Wilcoxon test (+VSP). (F) P2/P1 evoked I$_{Ca}$ and I$_{Ba}$ ratio for each cell. p-values determined by paired t-test (-VSP) and Wilcoxon test (+VSP).

the enzymatic activity of PLC [44, 45], Ca$^{2+}$ influx through Ca$_v$2.2 could amplify the effects of Gα$_q$ coupling to PLC, leading to greater reductions in PIP2 levels and reduced channel function than when using Ba$^{2+}$ as the permeant ion. The persistence of VDF in the presence of GDPβS could then be viewed as a disinhibition of Ca$_v$2.2 by stabilizing PIP2 levels that support channel function (Fig 8B). Some PLC isoforms are membrane-associated and can form macromolecular complexes with ion channels and GPCRs to allow for fast and localized signaling [46, 47]. Ca$_v$ channels interact with a variety of proteins [1] including those that may scaffold PLC and position it for regulation by incoming Ca$^{2+}$ ions. In addition, micromolar concentrations of Ca$^{2+}$ can cluster PIP2 in nanodomains [48, 49] which might make PIP2 a more appealing substrate for hydrolysis by PLC than in the presence of Ba$^{2+}$ ions.

PIP2 has complex actions on Ca$_v$2 channels, causing both a stimulation and inhibition of function [40]. According to one model, PIP2 binds to an "R" domain which, like G-proteins, causes channels to enter a "reluctant" (i.e., inhibited) mode of gating at intermediate voltages.

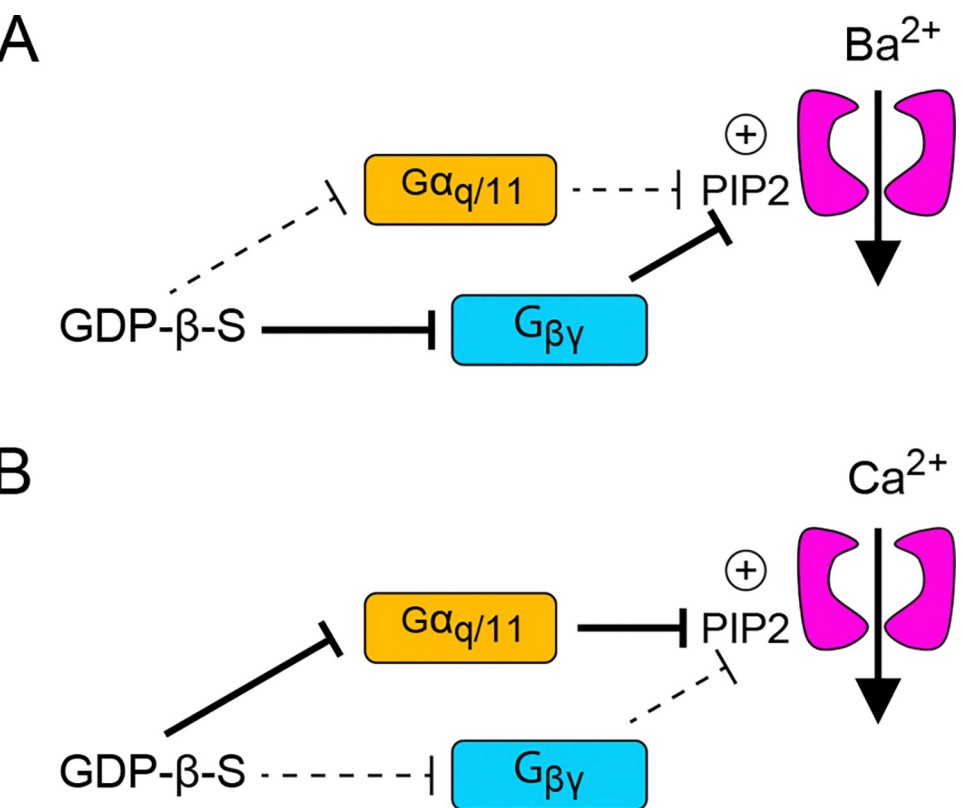

**Fig 8. Model for distinct G-protein modulation of I$_{Ba}$ and I$_{Ca}$ mediated by Ca$_v$2.2.** (A, B) Ca$_v$2.2 channels are potentiated by PIP2 binding and inhibited by Gβγ binding to the channel. Gα$_q$-mediated decreases in PIP2 would be expected to inhibit I$_{Ba}$ and I$_{Ca}$, whereas Gβγ would be expected to promote VDF. For I$_{Ba,}$ the main effect of GDP-β-S is to suppress VDF by inhibiting liberation of Gβγ from Gα$_{i/o}$ (A). For I$_{Ca}$, the main effect of GDP-β-S is to increase VDF by inhibiting Gα$_q$-mediated decreases in PIP2 (B).

In contrast, PIP2 binding to a stimulatory "S" domain is required for channel activation [40]. Structural and functional studies show that the S domain likely corresponds to a PIP2 binding site in domain II S4 [50–52]. Additionally, a second site in the cytoplasmic I-II linker is important for stimulatory effects of PIP2 on Ca$_v$2.2 channels containing the cytosolic β$_{2c}$ but not the membrane-tethered β$_{2a}$ subunit [52]. Apparently, the palmitoylation of β$_{2a}$ allows it to compete with PIP2 binding to the "S" domain, perhaps biasing its interaction with the "R" domain [52–54]. Compared to Ca$_v$2.2 channels with other β subunits, those containing β$_{2a}$ are less sensitive to the stimulatory effects, and more sensitive to the inhibitory effects, of PIP2; an increase in current density is seen upon Gα$_q$-linked receptor activation of β$_{2a}$-containing channels [53, 54]. The conversion of "reluctant" channels to "willing" channels upon Dr-VSP activation could contribute to the reduced VDF of $I_{Ca}$ if "willing" channels represent a "pre-facilitated" state. Based on this logic, the effects of Gα$_q$-mediated PIP2 depletion on VDF are expected to differ for Ca$_v$2.2 channels containing β$_{2a}$ vs. cytosolic β subunits (i.e., β$_{2c}$ or β$_3$), which may further diversify the modulatory properties of these channels in neurons. PIP2 has been shown to support VDF of Ca$_v$2.2 channels in hypothalamic neurons [55]. Moreover, Ca$^{2+}$ and PIP2 have been found to strengthen Gβγ-mediated inhibition of Ca$_v$2 channels [56–58]. Future studies are needed to dissect the mechanisms whereby alterations in PIP2 enable gating transitions that underlie VDF, and the interplay of Ca$^{2+}$, G-proteins, and Ca$_v$β subunits in this process.

## Supporting information

**S1 File. Datasets for Figs 1–7.** Datasets for analysis presented in Figs 1–7 of the article. (XLSX)

## Acknowledgments

The authors thank these individuals for gifts of cDNAs: Paul Kammermeier for MAS-GRK2-ct, Byung Chang Suh for the Dr-VSP, Diane Lipscombe for Ca$_v$2.2.

## Author Contributions

**Conceptualization:** Jessica R. Thomas, Amy Lee.

**Formal analysis:** Jessica R. Thomas, Jinglang Sun, Juan De la Rosa Vazquez, Amy Lee.

**Funding acquisition:** Amy Lee.

**Investigation:** Jessica R. Thomas, Jinglang Sun, Juan De la Rosa Vazquez.

**Project administration:** Amy Lee.

**Supervision:** Amy Lee.

**Writing – original draft:** Jessica R. Thomas.

**Writing – review & editing:** Jessica R. Thomas, Jinglang Sun, Juan De la Rosa Vazquez, Amy Lee.

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
