## [Decision Letter · Decision Letter 0]

27 Dec 2024

PONE-D-24-52981Complex regulation of Cav2.2 N-type Ca2+ channels by Ca2+ and G-proteinsPLOS ONE

Dear Dr. Lee,

Thank you for submitting your manuscript to PLOS ONE. After careful consideration, we feel that it has merit but does not fully meet PLOS ONE’s publication criteria as it currently stands. Therefore, we invite you to submit a revised version of the manuscript that addresses the points raised during the review process.

We look forward to receiving your revised manuscript.

Kind regards,

Steven Barnes

Academic Editor

PLOS ONE

Journal Requirements: When submitting your revision, we need you to address these additional requirements. 1. Please ensure that your manuscript meets PLOS ONE's style requirements, including those for file naming. The PLOS ONE style templates can be found at https://journals.plos.org/plosone/s/file?id=wjVg/PLOSOne_formatting_sample_main_body.pdf and https://journals.plos.org/plosone/s/file?id=ba62/PLOSOne_formatting_sample_title_authors_affiliations.pdf 2. We note that the grant information you provided in the ‘Funding Information’ and ‘Financial Disclosure’ sections do not match.  When you resubmit, please ensure that you provide the correct grant numbers for the awards you received for your study in the ‘Funding Information’ section. 3. We note that your Data Availability Statement is currently as follows: All relevant data are contained within the manuscript and its Supporting Information files Please confirm at this time whether or not your submission contains all raw data required to replicate the results of your study. Authors must share the “minimal data set” for their submission. PLOS defines the minimal data set to consist of the data required to replicate all study findings reported in the article, as well as related metadata and methods (https://journals.plos.org/plosone/s/data-availability#loc-minimal-data-set-definition). For example, authors should submit the following data: - The values behind the means, standard deviations and other measures reported;- The values used to build graphs;- The points extracted from images for analysis. Authors do not need to submit their entire data set if only a portion of the data was used in the reported study. If your submission does not contain these data, please either upload them as Supporting Information files or deposit them to a stable, public repository and provide us with the relevant URLs, DOIs, or accession numbers. For a list of recommended repositories, please see https://journals.plos.org/plosone/s/recommended-repositories. If there are ethical or legal restrictions on sharing a de-identified data set, please explain them in detail (e.g., data contain potentially sensitive information, data are owned by a third-party organization, etc.) and who has imposed them (e.g., an ethics committee). Please also provide contact information for a data access committee, ethics committee, or other institutional body to which data requests may be sent. If data are owned by a third party, please indicate how others may request data access. 4. Please review your reference list to ensure that it is complete and correct. If you have cited papers that have been retracted, please include the rationale for doing so in the manuscript text, or remove these references and replace them with relevant current references. Any changes to the reference list should be mentioned in the rebuttal letter that accompanies your revised manuscript. If you need to cite a retracted article, indicate the article’s retracted status in the References list and also include a citation and full reference for the retraction notice.

**Additional Editor Comments:**

Dear Dr. Lee,

These 2 reviews took roughly the normal time for completion.

I believe you will be able to respond effectively to both referee's concerns, which fall under Minor Revisions in my opinion.

Your emphasis would be best spent on the comments from Reviewer #2.

I look forward to your responses.

Best,

Steve Barnes

Reviewers' comments:

Reviewer's Responses to Questions

**Comments to the Author**

1. Is the manuscript technically sound, and do the data support the conclusions?

Reviewer #1: Yes

Reviewer #2: Partly

2. Has the statistical analysis been performed appropriately and rigorously? 

Reviewer #1: Yes

Reviewer #2: I Don't Know

3. Have the authors made all data underlying the findings in their manuscript fully available?

Reviewer #1: Yes

Reviewer #2: Yes

4. Is the manuscript presented in an intelligible fashion and written in standard English?

Reviewer #1: Yes

Reviewer #2: Yes

5. Review Comments to the Author

Reviewer #1: This study investigates how G-protein coupled receptors (GPCRs) inhibit Cav2.2 N-type Ca²⁺ channels through two distinct pathways: a fast, voltage-dependent pathway mediated by Gai/Gbg and a slow, voltage-independent pathway involving Gaq-dependent reductions in phosphatidylinositol 4,5-bisphosphate (PIP2) or increases in arachidonic acid. Unlike previous studies that primarily used Ba²⁺, this research compares the tonic G-protein inhibition of currents carried by Ba²⁺ (IBa) and Ca²⁺ (ICa). These results propose that when G-proteins are inhibited, Ca²⁺ influx through Cav2.2 promotes a form of VDF involving PIP2. The findings highlight the complexity of how Cav2.2 channels integrate G-protein signaling pathways, potentially enriching the information encoding capabilities of chemical synapses in the nervous system.

1. In Figure 3, why is the voltage-dependent facilitation (VDF) of ICa so strong, similar to that of IBa, while it appears much weaker than IBa in Figures 1 and 2? The differences should be clearly demonstrated.

2. To gain a deeper understanding of the molecular mechanism, it would be beneficial to compare the results of VDF experiments in channels with membrane-tethered β2a or cytosolic CaV β subunits, such as β1a or β3. I recommend testing the key findings in channels with cytosolic β subunits.

3. Please correct "β2A" to "β2a" in the first paragraph of the Materials and Methods and Results sections.

4. Kindly check the spacing between words, numbers, and units throughout the text.

Reviewer #2: I found the findings in this manuscript useful for advancing understanding of how lipids affect N-VGCC gating and modulation. The findings are timely considering there are now structural studies showing at least one PIP2 binding site to CaV2.2 (Gao et al, 2021; Dong et al 2021) as well as detailed biophysical data of channel movements from Nilsson et al. (2024).

File = Dropbox(personal)\\2024 Amy Lee Plos One ReviewF

Thomas et al have examined how tonic modulation of recombinant N-type calcium current by G-proteins varies depending on whether Ba2+ or Ca2+ is the permeant ion. Relatively few studies of N-current modulation have used physiological conditions where Ca2+ serves as the charge carrier. The authors raise the possibility that important Ca2+-dependent effects on channel gating might be lost when using Ba+ as the charge carrier. To test for differences with Ca2+ vs Ba2+ as the charge carrier, whole-cell currents were recorded from channels made up of CaV2.2-37b, α2δ-1 and �2a expressed in HEK293T cells. The authors used conditions known inhibit G-protein activity, by including in the pipette soluction GDP-�-S which blocks G-protein activity or transfection of HEK cells with the C-terminal construct of GPCR kinase (GRK), which has no kinase activity but acts to sequester Gβγ. They applied a series of classic voltage protocols to examine the voltage-dependent facilitation (VDF) of currents following a prepulse. The method is standard for looking at tonic inhibition of CaV2.2 by Gβγ.

Their results indicate that tonic current inhibition by Gβγ is context dependent. They found that tonic inhibition by Gβγ is stronger for IBa than for Ica as measured by differences in VDF. Moreover, their findings implicate PIP2 in VDF of ICa but not IBa when Gβγ-mediated inhibition is suppressed. While insensitive to high intracellular Ca2+ buffering, VDF of ICa that remained in cells dialyzed with GDP-β-S but was blunted by reductions in PIP2 when activating a voltage dependent phosphatase that acts on PI substrates. The authors concluded that when G-proteins are inhibited, Ca2+ influx through Cav2.2 promotes a form of VDF that involves PIP2. VDF of IBa2+ is minimally affected by VSP mediated metabolism of PIP2 levels. The findings reveal a complicated interplay of permeant ion, tonic G-protein activity and PIP2 levels that in part depend on the permeant ion.

The findings serve a useful purpose of pulling together previous observations about the complicated influences calcium, BAPTA, PIP2 and G-proteins exert on tonic CaV2.2 channel activity. The manuscript could contribute substantially to the calcium channel field if critical context in data interpretation on several issues is added by expanding discussion of several previous studies.

Context Issues:

1) CaV2 model of 2 PIP2 binding sites: The authors do not address previous findings that CaV2 channels appear to have 2 PIP2 binding sites that exert different effects on CaV2.2 activity. It may not appear obvious that their data may arise from both or just one PIP2 binding site. Previously, Wu et al (2002) presented a strong argument for two PIP2 binding sites on CaV2 channels: one site regulates whether channels open with “willing” voltage sensitivity (S site) and the second, lower affinity PIP2 binding site (R) promotes reluctant gating that is voltage-dependent where strong voltage pulses facilitate current amplitude to “willing” kinetics and amplitude. Loss of PIP2 from the R site enhances current amplitude promoting willing gating such that a prepulse no longer facilitates current amplitude. Thus, conditions that shift equilibrium in favor of PIP2 bound at the R site would be predicted to promote VDF following a prepulse. Loss of PIP2 at this site shifts the activation to more negative voltages and the channels undergo rapid activation and no VDF. Thomas et al’s should discuss their data in context of the Wu et al model since their data are complimentary to this model.

2) Palmitoylated �2a: The Hille, Rittenhouse, & Suh labs have each published findings that when �2a is expressed in HEK cells, its 2 palmitoyl groups, located on vicinal cysteines, appear occupy the “S” PIP2 binding site and confer resistance to current inhibition by M1R/Gq signaling and PIP2 breakdown. Under Thomas et al’s recording conditions, as with the above studies, the palmitoylated �2a will block the PIP2 “S” putative binding site leaving modulation of the voltage-dependent “R” site. The authors need to point this out since it is directly relevant to their data interpretation as mentioned above.

3) Previous expts with BAPTA & Ca2+: BAPTA can interact with PIP2 protecting it from breakdown. The authors need to look at Shapiro et al 1994 in Neuron, Rousset et al 2004a (JBC) and 2004b (FEBS Letters). Each of these studies address biophysical aspects of interactions among Ca2+, BAPTA, G-proteins. The data in these papers may help further mechanistic insight in the differences in VDF with Ba2+ vs Ca2+.

4) Alternative theories: while the authors conclude that PIP2 regulates ICa2+, rather than CaM kinases or G-protein mediated signaling cascades, the authors should not rule out a role for enzymes directly activated by Ca2+ and/or PIP2 without G�� activity. Good to cover all possibilities.

Major Issues

The prepulse voltage used varies with different figures. For example, Figs 1 &2 use a prepulse to +60 mV whereas Fig 7 it is +20 mV. The I-FF figs were apparently gathered from different prepulse voltages, e.g., Figs. 3F,5D, 6D plot I-FF80, whereas Figs 1G,2G & 4G, plot I-FF60, and Fig 7 data are plotted P2/P1 vs P2-P1/P1as with the other figures. It is not clear why in Fig. 7 the Pre pulse is only to +20 mV. The authors need to explain the thinking behind the different voltages and calculations. Explain why they used different voltages since it adds an additional variable when comparing experimental data.

For Fig. 7 include 2 graphs like those in Figs. 1C & F.

Activation kinetics of current traces were not measured despite it being a hallmark of G-protein modulation. The authors have the traces. The activation kinetics, e.g. time to peak or �o measurements should support changes in current amplitudes following prepulses. The extra data would strengthen their observation that Ca2+ vs Ba2+ current modulation show different characteristics.

Minor Issues

1. Everywhere Gi is written should be changed to Go/i. Indeed, Go may mediate more VDF than Gi.

2. State where the amplitude of current traces were measured, e.g., at the end of the test pulse, the peak tail current?

3. Explain why data are expressed as median rather than the mean.

4. The figures need more detail: what is the test pulse for all figs where current amplitudes and FF were calculated, e.g, for Fig 2 the test voltage was 0 mV but it is not clear for Fig. 2 C & F.

5. In Fig. 1, the IBa individual P1 traces are so small that the currents can’t be compared directly to the P2 traces. If possible, replace the recording for one where the current amplitudes are greater.

6. The I-FF data would be clearer if in Figs 1G, 2G, #F & 4G graphs use symbols other than circles, e.g. squares or triangles to distinguish the difference in data graphed compared to C & F.

6. PLOS authors have the option to publish the peer review history of their article (what does this mean?). If published, this will include your full peer review and any attached files.

Reviewer #1: No

Reviewer #2: No

---

## [Author Response · Author response to Decision Letter 0]

13 Jan 2025

We thank the reviewers for their careful review of our manuscript. We have addressed their concerns as outlined below:

Reviewer #1: 

1. In Figure 3, why is the voltage-dependent facilitation (VDF) of ICa so strong, similar to that of IBa, while it appears much weaker than IBa in Figures 1 and 2? The differences should be clearly demonstrated.

In Figs.1,2,4 the voltage protocol involves analyzing the effect of a +60 mV prepulse on currents evoked at different test voltages (P1, P2). Here, Inorm represents each test current normalized to the maximal current evoked by the second (P2) test pulse. In Fig.3,5,6 we analyze the effect of varying prepulses on test currents evoked by a single test voltage, before (P1) and after (P2) the prepulse. Here Inorm represents the P2 tail current divided by the P1 tail current. Therefore, the two protocols provide two different measures of VDF. To clarify the differences in these protocols, we modified the corresponding figure legends with these details.

2. To gain a deeper understanding of the molecular mechanism, it would be beneficial to compare the results of VDF experiments in channels with membrane-tethered β2a or cytosolic CaV β subunits, such as β1a or β3. I recommend testing the key findings in channels with cytosolic β subunits.

The focus of our study was on uncovering the influence of Ca2+ permeation on G-protein modulation of Cav2.2. Based on previous work (PMID: 36374183), we would expect that the difference in VDF of ICa and IBa would indeed depend strongly on the identity of the CaV β subunit. We agree that comparing results obtained with β1a or β3 with those we obtained with β2a would be interesting but feel that such experiments are beyond the scope of our study. Also in response to reviewer 2, we have re-written the last paragraph of the discussion which adds reference to comparative studies of Cavβ subunits as an important next step: 

 “PIP2 has complex actions on Cav2 channels, causing both a stimulation and inhibition of function [41]. According to one model, PIP2 binds to an “R” domain which, like G-proteins, causes channels to enter a “reluctant” (i.e., inhibited) mode of gating at intermediate voltages. In contrast, PIP2 binding to a stimulatory “S” domain is required for channel activation [41]. Structural and functional studies show that the S domain likely corresponds to a PIP2 binding site in domain II S4 [51-53]. Additionally, a second site in the cytoplasmic I-II linker is important for stimulatory effects of PIP2 on Cav2.2 channels containing the cytosolic β2c but not the membrane-tethered β2a subunit [53]. Apparently, the palmitoylation of β2a allows it to compete with PIP2 binding to the “S” domain, perhaps biasing its interaction with the “R” domain [53-55]. Compared to Cav2.2 channels with other β subunits, those containing β2a are less sensitive to the stimulatory effects, and more sensitive to the inhibitory effects, of PIP2; an increase in current density is seen upon Gαq-linked receptor activation of β2a-containing channels [54, 55]. The conversion of “reluctant” channels to “willing” channels upon Dr-VSP activation could contribute to the reduced VDF of ICa if “willing” channels represent a “pre-facilitated” state. Based on this logic, the effects of Gαq-mediated PIP2 depletion on VDF are expected to differ for Cav2.2 channels containing β2a vs. cytosolic β subunits (i.e., β2c or β3), which may further diversify the modulatory properties of these channels in neurons. PIP2 has been shown to support VDF of Cav2.2 channels in hypothalamic neurons [56]. Moreover, Ca2+ and PIP2 have been found to strengthen Gβγ-mediated inhibition of Cav2 channels [57-59]. Future studies are needed to dissect the mechanisms whereby alterations in PIP2 enable gating transitions that underlie VDF, and the interplay of Ca2+, G-proteins, and Cavβ subunits in this process.”

3. Please correct "β2A" to "β2a" in the first paragraph of the Materials and Methods and Results sections. Done

4. Kindly check the spacing between words, numbers, and units throughout the text. Done.

Reviewer #2

The manuscript could contribute substantially to the calcium channel field if critical context in data interpretation on several issues is added by expanding discussion of several previous studies.

Context Issues:

1) CaV2 model of 2 PIP2 binding sites: The authors do not address previous findings that CaV2 channels appear to have 2 PIP2 binding sites that exert different effects on CaV2.2 activity. It may not appear obvious that their data may arise from both or just one PIP2 binding site. Previously, Wu et al (2002) presented a strong argument for two PIP2 binding sites on CaV2 channels: one site regulates whether channels open with “willing” voltage sensitivity (S site) and the second, lower affinity PIP2 binding site (R) promotes reluctant gating that is voltage-dependent where strong voltage pulses facilitate current amplitude to “willing” kinetics and amplitude. Loss of PIP2 from the R site enhances current amplitude promoting willing gating such that a prepulse no longer facilitates current amplitude. Thus, conditions that shift equilibrium in favor of PIP2 bound at the R site would be predicted to promote VDF following a prepulse. Loss of PIP2 at this site shifts the activation to more negative voltages and the channels undergo rapid activation and no VDF. Thomas et al’s should discuss their data in context of the Wu et al model since their data are complimentary to this model.

2) Palmitoylated �2a: The Hille, Rittenhouse, & Suh labs have each published findings that when �2a is expressed in HEK cells, its 2 palmitoyl groups, located on vicinal cysteines, appear occupy the “S” PIP2 binding site and confer resistance to current inhibition by M1R/Gq signaling and PIP2 breakdown. Under Thomas et al’s recording conditions, as with the above studies, the palmitoylated �2a will block the PIP2 “S” putative binding site leaving modulation of the voltage-dependent “R” site. The authors need to point this out since it is directly relevant to their data interpretation as mentioned above.

Thank you for suggesting this line of discussion. In response to points 1 and 2, we modified the last paragraph of the discussion (see response to Reviewer 1, point 2).

3) Previous expts with BAPTA & Ca2+: BAPTA can interact with PIP2 protecting it from breakdown. The authors need to look at Shapiro et al 1994 in Neuron, Rousset et al 2004a (JBC) and 2004b (FEBS Letters). Each of these studies address biophysical aspects of interactions among Ca2+, BAPTA, G-proteins. The data in these papers may help further mechanistic insight in the differences in VDF with Ba2+ vs Ca2+.

We did not observe any effects of BAPTA (Fig.6B,D), which suggested that BAPTA was not acting to modify PIP2 levels relevant for VDF of ICa. Nevertheless, we appreciate the suggestion and incorporated citations of these references in the new last paragraph of the discussion.

4) Alternative theories: while the authors conclude that PIP2 regulates ICa2+, rather than CaM kinases or G-protein mediated signaling cascades, the authors should not rule out a role for enzymes directly activated by Ca2+ and/or PIP2 without G�� activity. Good to cover all possibilities.

This is an excellent point, and one that we already mentioned in the Discussion, paragraph 2: “Considering that Ca2+ can increase the enzymatic activity of PLC [45, 46], Ca2+ influx through Cav2.2 could amplify the effects of Gαq coupling to PLC, leading to greater reductions in PIP2 levels and reduced channel function than when using Ba2+ as the permeant ion.”

Major Issues

The prepulse voltage used varies with different figures. For example, Figs 1 &2 use a prepulse to +60 mV whereas Fig 7 it is +20 mV. The I-FF figs were apparently gathered from different prepulse voltages, e.g., Figs. 3F,5D, 6D plot I-FF80, whereas Figs 1G,2G & 4G, plot I-FF60, and Fig 7 data are plotted P2/P1 vs P2-P1/P1as with the other figures. It is not clear why in Fig. 7 the Pre pulse is only to +20 mV. The authors need to explain the thinking behind the different voltages and calculations. Explain why they used different voltages since it adds an additional variable when comparing experimental data.

We regret if this was unclear, but we explained the use of the different prepulse voltage in Fig.7 in the last paragraph of the Results section: “To enable VSP activation, we modified our voltage protocol to include a +120-mV VSP-activating pulse prior to the P1 test pulse and VDF was measured as the ratio of the P2/P1 pulses with an intervening +20 mV prepulse (Fig.7A). A more modest depolarizing prepulse was used in these experiments to avoid additional activation of the VSP.”

As described in our response to Reviewer 1, point 1, Fig.3,5,6 and Fig.1,2,4 utilize distinct voltage protocols to measure different aspects of VDF (pre-pulse dependence vs test-pulse dependence). As such, we believe the different metrics for VDF (FF80 vs FF60) are warranted, especially since we do not compare data obtained with different voltage protocols via statistical analysis.

For Fig. 7 include 2 graphs like those in Figs. 1C & F. Done.

Activation kinetics of current traces were not measured despite it being a hallmark of G-protein modulation. The authors have the traces. The activation kinetics, e.g. time to peak or �o measurements should support changes in current amplitudes following prepulses. The extra data would strengthen their observation that Ca2+ vs Ba2+ current modulation show different characteristics.

We respectfully chose not to perform this analysis since we expect the differences in kinetics of ICa and IBa to be rather subtle since both undergo kinetic slowing due to Gβγ- mediated inhibition, with faster onset following the depolarizing prepulse. Note the similar kinetics of ICa and IBa in Figs. 1,2. Moreover, the focus of our study was not on Gβγ- mediated inhibition, but rather how Gβγ- independent VDF is altered in Ca2+ vs Ba2+.

Minor Issues

1. Everywhere Gi is written should be changed to Go/i. Indeed, Go may mediate more VDF than Gi. Done.

2. State where the amplitude of current traces were measured, e.g., at the end of the test pulse, the peak tail current?

We clarified in the legends to Figs.1,2,4 that the current amplitudes were measured at the end of the 10-ms test pulse and that in Figs.3,5-7 that the tail current amplitudes were measured.

3. Explain why data are expressed as median rather than the mean.

The median was reported in cases where the data were non-normally distributed whereas the mean was reported when data were normally distributed. We added clarification to the legends for the statistically analyzed data as to whether the bar in graphs of grouped data represents the median or mean.

4. The figures need more detail: what is the test pulse for all figs where current amplitudes and FF were calculated, e.g, for Fig 2 the test voltage was 0 mV but it is not clear for Fig. 2 C & F. 

We have indicated that the test voltage was 0 mV in the corresponding figure legends.

5. In Fig. 1, the IBa individual P1 traces are so small that the currents can’t be compared directly to the P2 traces. If possible, replace the recording for one where the current amplitudes are greater.

Done.

6. The I-FF data would be clearer if in Figs 1G, 2G, #F & 4G graphs use symbols other than circles, e.g. squares or triangles to distinguish the difference in data graphed compared to C & F.

Done.

Additional changes:

--Added complete datasets from which figures were generated as Supporting Information

--Added Acknowledgements section

-- Added these references:

54. Suh BC, Kim DI, Falkenburger BH, Hille B. Membrane-localized beta-subunits alter the PIP2 regulation of high-voltage activated Ca2+ channels. Proc Natl Acad Sci U S A. 2012;109(8):3161-6. Epub 20120202. doi: 10.1073/pnas.1121434109. 

55. Heneghan JF, Mitra-Ganguli T, Stanish LF, Liu L, Zhao R, Rittenhouse AR. The Ca2+ channel beta subunit determines whether stimulation of Gq-coupled receptors enhances or inhibits N current. J Gen Physiol. 2009;134(5):369-84. doi: 10.1085/jgp.200910203. 

57. Shapiro MS, Wollmuth LP, Hille B. Angiotensin II inhibits calcium and M current channels in rat sympathetic neurons via G proteins. Neuron. 1994;12(6):1319-29. doi: 10.1016/0896-6273(94)90447-2. 

58. Rousset M, Cens T, Gouin-Charnet A, Scamps F, Charnet P. Ca2+ and phosphatidylinositol 4,5-bisphosphate stabilize a Gbeta gamma-sensitive state of Ca V2 Ca 2+ channels. J Biol Chem. 2004;279(15):14619-30. Epub 20040113. doi: 10.1074/jbc.M313284200. 

59. Rousset M, Cens T, Vanmau N, Charnet P. Ca2+-dependent interaction of BAPTA with phospholipids. FEBS Lett. 2004;576(1-2):41-5. doi: 10.1016/j.febslet.2004.08.058.

---

## [Editor Report · Decision Letter 1]

21 Jan 2025

Complex regulation of Cav2.2 N-type Ca2+ channels by Ca2+ and G-proteins

PONE-D-24-52981R1

Dear Dr. Lee,

We’re pleased to inform you that your manuscript has been judged scientifically suitable for publication and will be formally accepted for publication once it meets all outstanding technical requirements.

Kind regards,

Steven Barnes

Academic Editor

PLOS ONE

Additional Editor Comments (optional):

The authors provided excellent responses and changes to the manuscript based on the 2 reviewers thorough and insightful comments and requested corrections. This manuscript is now ready for publication.
---

## [Editor Report · Acceptance letter]

30 Jan 2025

PONE-D-24-52981R1 

PLOS ONE

Dear Dr. Lee, 

I'm pleased to inform you that your manuscript has been deemed suitable for publication in PLOS ONE. Congratulations! Your manuscript is now being handed over to our production team.

Kind regards, 

on behalf of

Dr. Steven Barnes 

Academic Editor

PLOS ONE